Microsaccades restore the visibility of minute foveal targets

Costela Francisco M. 1 2
McCamy Michael B. 1
Macknik Stephen L. 1 3
Otero-Millan Jorge 1 4
Martinez-Conde Susana 1 smart@neuralcorrelate.com
1 Department of Neurobiology, Barrow Neurological Institute , Phoenix, AZ , USA
2 Graduate Program in Neuroscience, Arizona State University , Tempe, AZ , USA
3 Department of Neurosurgery, Barrow Neurological Institute , Phoenix, AZ , USA
4 Department of Signal Theory and Communications, University of Vigo , Spain
Rocha Joao
Electronic publication date: 2013 Aug 1
Publication date: 2013
Volume: 1
Electronic Location ID: e119
Received 2013 May 29; Accepted 2013 Jul 10
Copyright: © 2013 Costela et al.
Copyright year: 2013
Copyright holder: Costela et al.
License: This is an open access article distributed under the terms of the Creative Commons Attribution License, which permits unrestricted use, distribution, and reproduction in any medium, provided the original author and source are credited.
License URL: https://creativecommons.org/licenses/by/3.0/

Keywords: Fading, Troxler fading, Fixation, Fovea, Illusion

Funding: National Science Foundation (Awards 0852636 1153786 to SMC and Award 0726113 to SLM) This study was supported by the Barrow Neurological Foundation (to SLM and SMC) and the National Science Foundation Awards 0852636 and 1153786 to SMC and Award 0726113 to SLM. The funders had no role in study design, data collection and analysis, decision to publish, or preparation of the manuscript.

==============================
Stationary targets can fade perceptually during steady visual fixation, a phenomenon known as Troxler fading. Recent research found that microsaccades—small, involuntary saccades produced during attempted fixation—can restore the visibility of faded targets, both in the visual periphery and in the fovea. Because the targets tested previously extended beyond the foveal area, however, the ability of microsaccades to restore the visibility of foveally-contained targets remains unclear. Here, subjects reported the visibility of low-to-moderate contrast targets contained entirely within the fovea during attempted fixation. The targets did not change physically, but their visibility varied intermittently during fixation, in an illusory fashion (i.e., foveal Troxler fading). Microsaccade rates increased significantly before the targets became visible, and decreased significantly before the targets faded, for a variety of target contrasts. These results support previous research linking microsaccade onsets to the visual restoration of peripheral and foveal targets, and extend the former conclusions to minute targets contained entirely within the fovea. Our findings suggest that the involuntary eye movements produced during attempted fixation do not always prevent fading—in either the fovea or the periphery—and that microsaccades can restore perception, when fading does occur. Therefore, microsaccades are relevant to human perception of foveal stimuli.

Introduction

During attempted fixation, the eyes are not still but continue to produce so called “fixational eye movements”, which include microsaccades, drift and tremor (Martinez-Conde, Macknik & Hubel, 2004). In the 1950s, several groups demonstrated that visual perception of stationary objects faded in the absence of eye movements (Ditchburn & Ginsborg, 1952; Riggs & Ratliff, 1952; Yarbus, 1957). Thus, fixational eye movements were linked to the prevention of visual fading and to the restoration of visibility during fixation, although the specific roles of each fixational eye movement type remained controversial. Microsaccades, the largest and fastest fixational eye movement, have since been connected to the visual restoration of faded targets (Martinez-Conde et al., 2006; McCamy et al., 2012), but their role in peripheral versus central vision has been less clear.

Recent research (McCamy et al., 2012) reported microsaccades to be the most important eye movement contributor to restoring faded targets, both in the visual periphery and in the fovea (the highest-resolution, rod-free region of the retina, subtending approx. 1.25 degrees of visual angle (Curcio et al., 1990)). The target size tested extended beyond the limits of the fovea, however, and so the ability of microsaccades to restore minute targets contained within the fovea continues to be in question.

Here, human subjects reported on the visibility of a target centered on the fovea and smaller than the area of the fovea, during attempted visual fixation. The target did not change physically, but its visibility decreased and increased intermittently during fixation, in an illusory fashion (a perceptual phenomenon known as Troxler fading (Troxler, 1804)). Microsaccade rates increased significantly before the target intensified, and decreased significantly before the target faded, for a variety of target contrasts. These results support previous research linking microsaccade onsets to the visual restoration of peripheral and foveal targets (Martinez-Conde et al., 2006; McCamy et al., 2012), and extend the former conclusions to small-size targets that are contained within the fovea.

Methods

Subjects

Eight subjects (7 males, 1 female) with normal or corrected-to-normal vision participated in the experiments. Six subjects were naive and were paid $15/session. Experiments were carried out under the guidelines of the Barrow Neurological Institute’s Institutional Review Board (protocol number 04BN039). Written informed consent was obtained from each subject.

Experimental design

Subjects rested their forehead and chin on the EyeLink 1000 head/chin support, ∼57 cm away from a linearized video monitor (Barco Reference Calibrator V, 75 Hz refresh rate). The experiment consisted of 4 sessions of ∼1 h, each including 50 randomly interleaved 30-second trials. The first session was counted as a training session and not included in the analyses.

While fixating a small red spot (0.03° diameter) on the center of the screen, subjects continuously reported whether a stimulus was faded/fading (button press) or intensified/intensifying (button release) (Martinez-Conde et al., 2006; McCamy et al., 2012; McCamy et al., 2013). To start the trial, subjects pressed a key and the stimulus appeared on the screen. Subjects were instructed to release the button as soon as they saw the stimulus. The stimulus was a two-lobe Gabor patch with a peak-to-trough width of 0.3° (Gaussian standard deviations of x = 0.2° and y = 0.2°; sine wave period of 1°; sine wave phase of 0). The Gabor was presented at the center of the screen and contained within the fovea, with contrast levels of 5%, 10%, 20%, and 40% from peak-to-trough and the same average luminance (50%) as the background. The orientation of the Gabor varied randomly between 0° and 360° in each trial, to control for orientation adaptation effects (Martinez-Conde et al., 2006; McCamy et al., 2012). After 30 s, the stimuli disappeared and the trial ended.

Eye movement analyses

Eye position was acquired noninvasively with a fast video-based eye tracker at 500 Hz (EyeLink 1000, SR Research). We recorded eye movements simultaneously in both eyes (instrument noise 0.01° RMS). We identified and removed blink periods as portions of the raw data where pupil information was missing. We also removed portions of data where very fast decreases and increases in pupil area occurred (>50 units/sample, such periods are probably semi-blinks where the pupil is never fully occluded) (Troncoso et al., 2008). We added 200 ms before and after each blink/semi-blink to eliminate the initial and final parts where the pupil was still partially occluded (Troncoso et al., 2008). Saccades were identified with a modified version of the algorithm developed by Engbert & Kliegl (Engbert, 2006; Engbert & Kliegl, 2003; Engbert & Mergenthaler, 2006; Laubrock, Engbert & Kliegl, 2005; Rolfs, Laubrock & Kliegl, 2006) with λ = 4 (used to obtain the velocity threshold) and a minimum saccadic duration of 6 ms. To reduce the amount of potential noise, we considered only binocular saccades, that is, saccades with a minimum overlap of one data sample in both eyes (Engbert, 2006; Engbert & Mergenthaler, 2006; Laubrock, Engbert & Kliegl, 2005; Rolfs, Laubrock & Kliegl, 2006). Additionally, we imposed a minimum intersaccadic interval of 20 ms so that potential overshoot corrections might not be categorized as new saccades (Moller et al., 2002). Microsaccades were defined as saccades with magnitude <1° in both eyes (Martinez-Conde et al., 2009; Martinez-Conde, Otero-Millan & Macknik, 2013). To calculate microsaccade properties such as magnitude and peak velocity we averaged the values for the right and left eyes. Figs. 1A–1B shows the magnitude distribution (Fig. 1A) and peak velocity-magnitude relationship (Fig. 1B) for both microsaccades (<1°) and saccades (≥1°). All subsequent analyses (Figs. 3–5) concern microsaccades only, that is, saccades with magnitude <1°.

Figure 1 Descriptive statistics for microsaccades (<1°) and saccades (≥1°).

(A) Microsaccadic and saccadic magnitude distribution for all subjects combined (n = 8). (B) Microsaccadic and saccadic peak velocity–magnitude relationship for all subjects combined. Each orange dot represents a microsaccade or a saccade with peak velocity indicated on the y-axis and magnitude indicated on the x-axis. The inset legend shows the microsaccade descriptive statistics. Error bars and shadows indicate the SEM across subjects.

Microsaccade correlations with reported transitions

Let XM and XR be the stochastic processes representing the onsets of microsaccades and intensification reports. For example, if s1, s2, …, sk are the start times of all the microsaccades for a given subject, then XM for that subject will be given by XM(t) = 1 if t = si for some 1 ≤ i ≤ k, and XM(t) = 0 otherwise; similarly for intensification reports.

We obtained correlations of microsaccades with reports of intensification for each subject, using ξMR(t)=∑n=−∞n=∞XM(n+t)XR(n) and then converting it to a rate (similarly for transitions to fading) (McCamy et al., 2012). For each subject, correlations were smoothed using a Savitzky-Golay filter of order 1 and a window size of 151 ms (Martinez-Conde et al., 2006). Average correlations are the average of the smoothed correlations (Figs. 3–4).

ROC analysis

We used a sliding receiver operating characteristic (ROC) analysis (Britten et al., 1992; Feierstein et al., 2006; Green & Swets, 1966; Hernandez, Zainos & Romo, 2002; Romo, Hernandez & Zainos, 2004; Romo et al., 2002; Troncoso et al., 2008) to quantify how well microsaccade rate may predict the type of perceptual transition (towards intensification versus fading) reported by the subjects, for the different foveal target contrasts. The area under an ROC curve provides a measure of the discriminability of two signals and is directly related to the overlap of the two distributions of responses that are compared (Feierstein et al., 2006). In our case, the area under the ROC curve can be interpreted as the probability with which an ideal observer, given the microsaccade rate during a window of time around a particular transition, can correctly determine the type of transition (towards fading or intensification). An ROC area of 0.5 corresponds to completely overlapping distributions (the ideal observer cannot discriminate between the two types of transitions); an area of 1 corresponds to transitions that can be perfectly discriminated based on microsaccade rate. This analysis makes no assumptions about the underlying distributions (Feierstein et al., 2006). For a given point in time, we compared the microsaccade rate distributions for transitions to intensification (true-positive rate) and transitions to fading (false-positive rate) for each subject. To obtain the ROC curve at that time, we plotted the probability of true positives as a function of the probability of false positives for all possible criterion response levels. We performed a sliding ROC analysis (kernel width 500 ms, slid in 2-ms increments) to calculate each subject’s area under the ROC curve at each time point around the transition. To determine the time point at which the ideal observer became better than chance, we calculated significance using a permutation procedure (Feierstein et al., 2006; Hernandez, Zainos & Romo, 2002; Romo, Hernandez & Zainos, 2004; Romo et al., 2002; Siegel & Castellan, 1988; Troncoso et al., 2008) with n = 1000 shuffles for each subject and a criterion p-value <0.01.

Statistical methods

To analyze the effect of target contrast on time faded per trial and rate of fading onsets (Figs. 2A–2B), we conducted separate single-factor repeated measures ANOVAs with the different contrast levels tested as the within-subjects factor. All post-hoc comparisons were done using Tukey’s HSD method. To assess whether microsaccade rates before transitions to intensification were significantly higher than those before transitions to fading, we performed one-tailed paired t-tests in each bin (bin size = 20 ms), using Bonferroni correction to account for multiple comparisons (Figs. 3–4). Significance levels were set to α = 0.01 throughout.

Figure 2 Perceptual reports.

(A) Average time faded per trial for each target contrast. The time faded per trial decreased linearly with target contrast (F(3, 21) = 88.48, p < 0.001; linear trend F(1, 7) = 112.36, p < 0.0001). (B) Fading onset rate for each target contrast. The effect of contrast was significant (F(3, 21) = 8.58, p = 0.0065). A Tukey HSD posthoc comparison showed a significant difference only between the 10% and 40% contrast target (p < 0.01). (C)–(F) Distribution of the durations of intensification and fading periods for each target contrast. Error bars and red and blue shadows indicate the SEM across subjects (n = 8).

Figure 3 Microsaccade correlations with reported transitions.

(A)–(D) Percent increase in microsaccade rate over baseline (i.e., relative to the average microsaccade rate (dashed horizontal line) for a given target contrast) around reported transitions toward intensification versus fading, for each target contrast. The solid vertical line indicates the reported transitions (t = 0). Target contrast is indicated at the top of each panel. The gray lines at the top indicate the bins where microsaccade rates before transitions to intensification were significantly higher than microsaccade rates before transitions to fading (see Methods for details). Red and blue shadows indicate the SEM across subjects (n = 8).

Figure 4 Correlations between microsaccades of different sizes and reported transitions.

Microsaccade sizes (0–15 arcmin; 15–30 arcmin; 30–45 arcmin; 45–60 arcmin) are indicated at the top of each panel and we have collapsed across target contrasts. All other details as in Fig. 3.

Figure 5 ROC analysis.

(A)–(C) The ideal observer can predict the type of illusory transitions (intensification vs. fading) based on microsaccade rate. The green line is the area under the ROC curve at any given time. The solid horizontal gray line indicates the significance level (i.e., the level at which the ideal observer performs above chance (horizontal dashed line; see Methods for details). Target contrast is indicated at the top of each panel. Significance is reached whenever the green line is above the grey line. The ideal observer’s prediction (green line) becomes significantly better than chance ≈ 800 ms before the reported transitions, for target contrasts of 10%, 20% and 40%. The shaded green area indicates the SEM across subjects (n = 8).

Results

Subjects fixated a small spot on the center of a computer screen and continuously reported, via button press/release, whether an unchanging visual target (a two-lobe Gabor patch with 5%, 10%, 20%, or 40% contrast), presented within the fovea, was faded (or in the process of fading) versus intensified (or intensifying) (Martinez-Conde et al., 2006; McCamy et al., 2012). Fading prevalence decreased as the target’s contrast increased (Fig. 2A), and the 10% contrast target generated the largest number of perceptual transitions, as indicated by subjects’ reports (Fig. 2B). As expected, lower-contrast targets were faded for longer time periods than higher contrast targets; thus the 5% contrast target resulted in the longest fading periods and the 40% contrast target in the longest intensification periods, whereas the 10% contrast target produced fading and intensification periods of comparable length (Figs. 2C–2F).

Microsaccade rates increased before transitions to intensification and decreased before transitions to fading, with the intermediate-contrast targets (10% and 20%) showing the strongest correlations between microsaccade rate increases and intensification reports (Fig. 3).

Microsaccade magnitude

We analyzed the effects of microsaccade size on perceptual transitions to intensification and fading. To do this, we separated all microsaccades in 4 different categories according to size (0–15 arcmin, 15–30 arcmin, 30–45 arcmin, and 45–60 arcmin) and correlated them to the perceptual intensification and fading reports for each target contrast (Fig. 4). The smallest microsaccades failed to restore target visibility, especially for those targets with the lowest levels of contrast (not shown). This result is consistent with the previous finding that larger microsaccades are more efficacious than smaller ones, possibly due to their increased ability to bring the neuronal receptive fields to uncorrelated stimulus regions (McCamy et al., 2012). As microsaccades grew in size, their correlation with perceptual transitions became stronger, also consistent with previous research (McCamy et al., 2012).

Receiver operating characteristic (ROC) analysis

To further quantify our conclusions, we conducted a sliding ROC analysis to calculate the ability of an ideal observer to predict the type of perceptual transition (towards intensification or fading) based on microsaccade rates. Figure 5 shows that the ideal observer becomes significantly better than chance (determined by permutation analysis; see Methods for details) ≈800 ms before the reported transitions, for targets of 10%, 20%, and 40% contrast.

Discussion

In 1804, Troxler described the perceptual fading of stationary objects during fixation, a perceptual phenomenon that came to be known as Troxler fading (Troxler, 1804). Despite Troxler’s report that not only peripheral, but also centrally fixated targets were susceptible to fading, Troxler fading became equated with peripheral fading in subsequent decades (see Wade & Tatler (2005) for a historical review). Yet, foveal fading has been reported by numerous researchers in a number of experimental conditions (Darwin, 1794; Krauskopf, 1963; McCamy et al., 2012; Pessoa & De Weerd, 2003; Simons et al., 2006; Troxler, 1804). A recent study found that microsaccades counteracted the perceptual (i.e., Troxler) fading of peripherally and foveally presented Gabor patches with peak-to-trough widths of 2.5° (McCamy et al., 2012). Because the edges of such centrally presented targets extended beyond the foveal limits, here we asked whether smaller size targets, constrained to the area of the fovea, might similarly fade from perception, and if so, whether microsaccades could restore their visibility as well.

Subjects reported the visibility of centrally presented two-lobe Gabor patches with peak-to-trough widths of 0.3° (that is, 8 times smaller than in McCamy et al. (2012)), of varying contrasts (5%, 10%, 20%, 40%). As with previous fading experiments (Martinez-Conde et al., 2006; McCamy et al., 2012; Spillmann & Kurtenbach, 1992; Troncoso et al., 2008), subjects reported that the perceptual state of the (foveally contained) targets appeared to oscillate between the faded/fading state and the visible/intensifying state. Thus, minute targets constrained to the fovea are subject to perceptual fading during fixation, in much the same manner as larger, foveally centered targets, and peripherally presented targets are (Martinez-Conde et al., 2006; McCamy et al., 2012). Microsaccade rates increased before transitions to visibility and decreased before transitions to invisibility, also in agreement with previous reports (Martinez-Conde et al., 2006; McCamy et al., 2012; Troncoso et al., 2008). These findings indicate that microsaccades can restore visibility across the retina, for a variety of target sizes and contrasts.

Target contrast and foveal fading

As one might have expected, lower-contrast targets faded more often and for longer periods of time than higher-contrast targets (Fig. 2). Perhaps more surprisingly, targets of moderate contrast levels (i.e., 20% and 40%) also faded, albeit less often, and microsaccades restored them perceptually at such times (Figs. 2–3 and 5). The correlation between microsaccade production and visual restoration of faded targets was most obvious for the intermediate contrast targets (i.e., 10% and 20%), although still present for lower and higher contrasts (i.e., 5% and 40%) (Figs. 3 and 5). Why did microsaccades restore the visibility of low-contrast (i.e., 5%) targets less effectively than that of intermediate-contrast targets? It seems likely that, when target visibility is highly degraded (for instance, due to minimum contrast levels), shifting of the retinal image due to microsaccades or other eye movements may not produce enough photoreceptor stimulation to generate a perceptual experience in a reliable way. Conversely, when intermediate or moderate contrast targets fade from perception, their visibility may be degraded just enough that microsaccades are able to restore them with high efficiency.

The current results are consistent with previous evidence that stimuli do fade at all retinal eccentricities (Wade & Tatler, 2005), despite the concerted actions of the three types of fixational eye movements (i.e., microsaccades, drift and tremor), and that microsaccades can successfully bring back such faded stimuli (McCamy et al., 2012).

Are microsaccades relevant to foveal visibility?

The present results show that microsaccades can restore the visibility of small, foveally contained targets of low-to-moderate (i.e., ranging from approximately 40% to 10%) contrasts. Very low contrast targets (i.e., 5% contrast) were visible for a small amount of the time only, and microsaccades did not significantly restore their visibility (although there was a trend between microsaccade production and target intensification (Figs. 3–5), especially for the larger microsaccade sizes (not shown)). High-contrast targets (i.e., higher than 40% contrast) were not tested in this study, but the current data suggest that they would have faded too briefly and rarely for microsaccades to restore their visibility in a substantial manner.

In light of these combined results, what is the value of microsaccades to foveal perception? We posit that—despite assumptions that only high-contrast stimuli are pertinent to foveal vision—most visible stimuli are relevant to perception by definition, regardless of their contrast. That is, fading of any stimulus (i.e., irrespective of contrast) is a visual degradation that microsaccades often supersede. Indeed, there are many small-sized, low-contrast objects that one might want to see optimally with central vision. A diamond earring on a white carpet, or small features in medical and research images, are two of many examples. Thus, the fovea has the capacity to inspect stimuli of all contrasts and spatial frequencies, and microsaccades can restore the visibility of a range of such foveal stimuli, after fading sets in. Therefore, microsaccades are relevant to human perception of foveal stimuli.

Future research should investigate the ability of microsaccades to restore faded targets of diverse spatial frequency content, at varied retinal eccentricities.

On the visibility of fixation targets

One frequent argument against the value of microsaccades to foveal vision is that fixation targets, which are presented centrally, never fade, even in the absence of microsaccades. We note that the sizes, colors and shapes of fixation targets used in vision studies vary widely (McCamy et al., 2013; Thaler et al., 2013), but their contrasts (and high spatial frequency content) are almost universally maximized. The present results suggest that fixation targets remain perpetually visible by virtue of their high contrast (and possibly high spatial frequency), rather than their small size and/or foveal presentation. Further, our results show that it is possible for one foveal stimulus to fade from perception (i.e., the low-to-moderate contrast Gabor patches presented here) and for another foveal stimulus to remain visible (i.e., the fixation target), at the same time.

Complete fading versus partial loss of visibility during fixation

Troxler fading, the perceptual experience at the heart of the present study, is a gradual, rather than an instantaneous process. Often, an object becomes less and less visible until it eventually disappears (and then reappears, typically when microsaccades bring it back, as shown here). Other times, an object’s visibility decreases at first, and then it is restored (again, usually in connection with microsaccade production) before complete fading has occurred. The current research set out to quantify the precise timing of the interactions between microsaccades and perceptual experience (similarly to Martinez-Conde et al. (2006) and McCamy et al. (2012)). Thus, experimental subjects indicated when the target was faded/fading versus intensified/intensifying, rather than reported merely when the target was completely faded versus fully visible. Had we considered only “total fading” and disregarded “partial fading” events, we would have achieved an incomplete picture of the role of microsaccades in visual restoration, rather than the full, dynamic picture of the interactions between microsaccade production and the ongoing perceptual experience that characterizes natural vision. Future research may investigate how microsaccades and other fixational eye movements impact the perception of gradations in fading/visibility (i.e., by obtaining a continuous measure of the subject’s perceptual experience, as in Simons et al. (2006)), rather than focus on the perceptual transitions to increased or decreased visibility (i.e., as in the present paradigm).

Fading prevention versus visibility restoration

Fixational eye movements are thought to overcome loss of vision by thwarting the neural adaptation (and thus the visual fading) ensuing from stable stimulation of the retinal receptors (Martinez-Conde, Macknik & Hubel, 2004). A fruitful discussion of the perceptual effects of microsaccades—in central vision and at other retinal eccentricities—must separately address their impact on counteracting (that is, reversing) fading versus preventing the fading from occurring in the first place (Martinez-Conde, Otero-Millan & Macknik, 2013). Here we set out to address the ability of microsaccades to counteract (i.e., reverse) fading; that is, to restore the visibility of already faded objects. Future research should establish the ability of the different types of fixational eye movements to prevent rather than counteract fading (i.e., to prevent vision loss rather than restore faded vision), as well as the physiological mechanisms by which fixational eye movements prevent and counteract neural adaptation.

Previous studies found that drift does not contribute strongly to reversing fading (Martinez-Conde et al., 2006; McCamy et al., 2012). Whereas microsaccades counteract fading once it has occurred, it is possible that both microsaccades and drift work together to prevent fading before it happens. Future research should also investigate this hypothesis.

In sum, fixational eye movements serve to prevent fading in the fovea and elsewhere, but not perfectly. Microsaccades have the ability to bring stimuli back to perception, when peripheral and foveal fading do occur.

We thank Behrooz Kousari for technical assistance.

Additional Information and Declarations

Competing Interests

Author Contributions

Human Ethics

Susana Martinez-Conde and Stephen L. Macknik are Academic Editors for PeerJ.

Francisco M. Costela conceived and designed the experiments, performed the experiments, analyzed the data, wrote the paper.

Michael B. McCamy conceived and designed the experiments, analyzed the data.

Stephen L. Macknik and Jorge Otero-Millan conceived and designed the experiments.

Susana Martinez-Conde conceived and designed the experiments, wrote the paper.

The following information was supplied relating to ethical approvals (i.e., approving body and any reference numbers):

Barrow Neurological Institute’s Institutional Review Board (protocol number 04BN039).

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
