# Peer review of "Microsaccades restore the visibility of minute foveal targets"

_PeerJ, doi:10.7717/peerj.119_

## Round 0.1 · original submission · Minor Revisions

Reviewer 1 has asked for additional discussion about why low-contrast stimulation failed to be restored bymicrossacades and the mechanisms that could be involved in the restoration of the foveal target visibility after microsaccades.

Reviewer 1 ·

Basic reporting

No comments.

Experimental design

No Comments

Validity of the findings

No Comments

Additional comments

The manuscript aimed to investigated a paradigm that previously showed that stationary objects faded without eye movements. The new contribution of the present manuscript was to focus that investigation in the foveal vision. They had interesting results that showed that microsaccades restored the visibility of foveal targets. They showed that the higher stimulus contrast, the higher microsaccade rate. The discussion is very good, but I think they can stress more why low-contrast stimulation failed to be restored by microssacades. What kind of mechanisms would be involved in the restoration of the foveal target visibility after microsaccades? The method used by the authors actually could be used to estimate the contrast sensitivity of that possible mechanism.

Reviewer 2 ·

Basic reporting

The submission adheres to all PeerJ policies.

Experimental design

The experimental design is appropriate and it has been thoroughly explained. The statistical analysis is rigorous including the correction for multiple comparisons.

Validity of the findings

The findings are not only valid but they provide a much needed complement to a previous report by the same authors suggesting that small saccades produced during attempted fixation can restore visibility of faded targets at all eccentricities, including the fovea.

---

## Round 0.2 · accepted · Accept

Dear Dr Susana Martinez-Conde,
I am pleased to inform that your manuscript has now been accepted for publication in Peer J.